# Plant Use in the Late Renaissance Gardens of the 17–18th Century Transylvania

**DOI:** 10.3390/plants12091798

**Published:** 2023-04-27

**Authors:** Albert Fekete, Máté Sárospataki

**Affiliations:** Department of Garden Art and Landscape Design, Institute of Landscape Architecture, Urban Planning and Garden Art, Hungarian University of Agriculture and Life Sciences, 1118 Budapest, Hungary; sarospataki.mate@uni-mate.hu

**Keywords:** renaissance garden art, plant use, landscape architecture, historic garden, garden heritage

## Abstract

The aim of this article is to find, scientifically define, and locate the most frequent occurrences of the gardens of Transylvania in the Late Renaissance period (17–18th centuries), and to collect and prepare a comprehensive plant list of these gardens. During our investigation, based on archival and literary sources, as well as field studies carried out, we identified 81 Late Renaissance residency gardens located in Transylvania. We defined the most typical garden types for the region and we delineated the most characteristic ornamental, fruit, and vegetable plants, including fodder plants, used at that time in residential gardens. Meanwhile, the article intends to give a general overview of the first decisive time period in the Carpathian Basin, represented by the Late Renaissance garden art, from a garden and landscape architectural point of view.

## 1. Introduction, Historical Background

The use of plants in Renaissance gardens is the subject of numerous studies and essays on botanical history in Europe [1,2,3,4,5,6,7,8,9,10,11,12,13,14,15,16,17,18,19,20,21,22]. In Hungary, there are also some historical and contemporary works on Renaissance Garden culture and the plants used [23,24,25,26,27,28,29,30,31,32], but research on the subject is far from what is desirable in relation to its importance. Research on Renaissance gardens in Transylvania has begun [23,24,25,26,27,28,29,30,31,32,33,34,35,36,37,38,39,40,41,42], but the amount of material on the plant population of gardens is small and tangential.

The study of the cultivated plants of the Transylvanian Renaissance gardens can provide interesting contributions to the history of gardens both locally and universally, since while in most countries outside Italy the Renaissance idea spread only at the beginning of the 16th century, the style appeared very early in the Carpathian Basin, and thus in Hungary, at around 1470. The launch of the style was underpinned by Hungary’s strong political, dynastic, and cultural ties with Italy, the dominant factor of which was the marriage of King Matthias to Aragonian Beatrix in 1474. What followed as a direct consequence of the matrimony was the influx of notable Italian painters, sculptors, and architects of the early Renaissance to the Hungarian Royal Court [43,44]. 

This first, early period of the Renaissance in the Carpathian Basin was related to the royal court and its immediate surroundings, and it lasted until the death of King Matthias in 1490. Although the death of the patron King Matthias in 1490 had a negative impact on the development of Renaissance ideas and art in Hungary, the century and a half of Ottoman occupation that followed the defeat of Mohács in 1526 caused a much more serious cultural disruption. During this period, a significant part of the country’s territory came under Ottoman rule, isolated from Europe.

The period between the end of the 16th century and the first decades of the 18th century is known as the second period of the Transylvanian Renaissance, which is considered the late Renaissance by art history. Due to the isolation and different cultural impacts caused by the Ottoman occupation, special forms of local characteristics developed in this period, making the historic garden features particularly rich and interesting in Transylvania. And although the Transylvanian Late Renaissance lasts for almost two centuries, its stylistic characteristics are most strongly expressed in the 17th century, which can be explained by the economic boom of the region. The independent Principality of Transylvania was in its golden age during the 17th century, when Gábor Bethlen, György Rákóczi I. and Mihály Apafi I. were the ones who financed the flourishing period of the country, also having a strong positive influence on the development of the garden culture of the region, especially in the case of residential gardens.

Although a few manor gardens are known from this period—among others the gardens of Mihály Apafi and his wife Anna Bornemisza at Ebesfalva (nowadays Erzsébetváros/Dumbraveni, RO), Küküllővár (Cetatea de Balta, RO), Székelyhíd (Sacueni, RO), Radnót (Iernut, RO) and Fogaras (Fagaras, RO) were famous in their time—the late Renaissance Transylvanian garden culture of the 17th and 18th centuries is a less researched topic in landscape architecture, the exploration of which is also helped by the research and systematic assessment of archival data on the plant use of gardens of the period.

## 2. Objectives

The main objective of the research is the thorough and systematic literature review of the Late Renaissance (17–18th century) garden culture in Transylvania, the synthesis and supplementation of the contemporary garden history of the region, based on archival sources. The study and analysis of plant species used in gardens of the period is a separate research topic within the history of gardens. The comparative analysis of the cultivated plants (ornamental plants and crops) of the manor gardens, going back four hundred years, provides new data and findings to the garden art of Transylvania in the 17th and 18th centuries, while at the same time placing Transylvanian gardens in a European context.

## 3. Research Methodology

Our research regarding plant use in Transylvanian late renaissance gardens started in 2016, as part of a comprehensive investigation of residential gardens from Transylvania. Our approach is based on the principles of case study research. Accordingly, each site is considered as a case study and analyzed separately before a comparison is made. In the analysis, we used an explicit analytical framework in order to compare different sites with different geographical, economical, and architectural contexts by different owners. Methods of data collection: comprising first of all a quantitative investigation of the existing archival (primary and secondary) sources and materials resulting in a first overview per case. As primary sources, we have to mention here the Archives of the HungarianAcademy of Science (Magyar Tudományos Akadémia Levéltára), and the county branches of the Romanian State Archives (Arhivele Nationale Romane, Filiala Cluj Napoca; Serviciul Judetean Mures al Arhivelor Nationale). Other archives, libraries, collections, etc. were used as well during the material collection.

The research was conducted in three phases: Identification of all Renaissance gardens in the study area, by examining and mapping their spatial/geographic location.Definition of three fundamental types, based on the data of the study sites
Type A: sites where the garden is not only mentioned, but described specifically with its parameters;Type B: sites where the garden is just mentioned, one or more gardens exist, but no description of its layout can be found.Investigation and analysis of Type A sites, by the cultivated plant types and species (ornamental plants, vegetables, fruits, herbs, agricultural crops, fodder plants), based on archival materials as follows:
the research of the distinct, clearly separable garden units defined by different types of cultivated plants of the era;the analysis of the frequency of the most typical plant species. 

### 3.1. Identification of All Renaissance Gardens Sites in the Study Area

The study area is Transylvania, a region of the Carpathian Basin located in Eastern Europe, which is now part of Romania. During the archival research, 86 sites were identified where there was a Late Renaissance Garden existing. The mapping of the identified sites clearly shows their spatial location within Transylvania (Figure 1a,b). The names of the locations can be identified using the reference numbers form in Table 1.

### 3.2. Definition of the Fundamental Garden Types, Based on the Data of the Study Sites 

Out of the 86 locations identified, in 50 cases, archive documents describe in detail the existence of the garden, its units, elements, and plants (“Type A”). In the remaining 34 cases, the garden is only mentioned, i.e., the existence of one or more gardens is referred to, but no specific description of them is to be found (“Type B”). (Figure 1b and Table 1.)

### 3.3. Investigation and Analysis of “Type A” Sites, by the Cultivated Plant Types and Species

This section contains the source research of the plant material of the gardens. Source research on garden history is usually based on three major sources of data: written, pictorial, and material memories.

For over more than 20 years, Transylvanian castle-garden ensembles, including Renaissance gardens, have been investigated, described, and analyzed by a research group from Hungary, led by Albert Fekete. The goal of this study of Transylvanian ensembles is to obtain background information for developing a strategy of landscape preservation and development in the long run that comprises the cultural and historical values and the demands from society on what to do with them in the contemporary context. Site visits and surveys represented very important pillars of the investigation, documenting the current condition of these historic ensembles.

As far as Transylvanian Late Renaissance Garden art is concerned, the material heritage is rather scarce. None of the Transylvanian castle gardens survive in their original form, and very few elements of Renaissance garden art have survived. 

What has survived is mainly associated with large-scale earthworks, such as those related to water features (canals, moats, fishponds) and terraces (e.g., orchards on hillsides). The destruction or transformation of gardens can be attributed to two main causes. One is the change in the style of gardens, with late Renaissance gardens being radically transformed in the 18th century according to Baroque and then in the 19th century to Landscape Garden ideas in all locations. Another reason for the deterioration is that the extremely fragile, mostly herbaceous vegetation, which was the typical material and essence of Late Renaissance gardens, has deteriorated considerably and disappeared completely in the absence of regular and professional maintenance.

The pictorial sources are also poor, and apart from a few illustrations in books on medical botany, there are only a few concrete details of plant use. 

In the absence of material and pictographic references, the present research is based on written sources, based on the understanding of the text, among which the unpublished works, usually preserved in archives, and works of scientific nature are of particular importance.

The written sources of garden history are diverse: from documents containing seemingly insignificant mentions and poor data to detailed garden descriptions, they contain a wide range of authentic information on garden history. Garden descriptions from primary and secondary sources often provide scientific detail on contemporary garden structures, gardening techniques, and plant species used in gardens. Written sources are generally grouped into the following categories:oInventories, registers, and fief ownership chartersoAccounts, payrolls, and expenditure certificatesoCorrespondenceoChronicles, traveloguesoDiaries, diary fragments, reminiscencesoWritings on local history, monographsoScientific and professional literature, dissertations, summariesoJournals, periodicals, newspapers, and other news items

Of these, the most relevant written sources for the study of late-Renaissance Transylvanian garden art are inventories and registers. In this work, we rely on the analysis of these sources.

Inventories and registers are primarily records of economic importance, which give detailed descriptions or more concise mentions of the gardens and estates of noble families. They provide a rich written source material for the history of Transylvanian noble gardens of the 16th and 18th centuries, and the plant material used. The value of these economically important records as sources for the history of art has been pointed out by several researchers, of whom Margit B. Nagy [33,45,46], Jolán Balogh [47,48,49,50,51], and Zsigmond Jakó [52,53,54,55] are of particular importance with regard to the Transylvanian aspects.

The importance of the documents on economic history for garden history is due to the fact that not only ornamental gardens, parks associated with castles and understood as artistic compositions, but also various types of kitchen gardens, such as vegetable gardens, orchards or vineyards, are an integral part of our castle garden culture. Until the 17th century, not only in Transylvania, but also in European garden culture, there was no sharp distinction between ornamental gardens and kitchen gardens. Garden inventories provide information mainly on the crops and ornamental plants grown and the gardening techniques used. However, they also contain valuable references to the structures and artistic qualities of the gardens.

By way of example, here are three brief passages illustrating the nature, content and detail of the written sources analyzed:

According to the 1681 garden inventory of the Kemény Castle in Nagysajó (Sieu, BN), the castle had several gardens, and in one of the “more beautiful” gardens, in raised beds, various crops and ornamental plants were grown:

“Rectangular beds enclosed by beams, with very pretty vines growing on the edges of the beds; again, rose and lilac bushes. Lots of fragrant grasses and flowers and some small crops. Again there are other compartments, without fences, in which under the fruit trees there are lilies of the valley.”[56]

Extract from the 1688 inventory of the vegetable garden of the Béldi Manor Garden in Csíkkozmás (Cozmeni, RO):

“There are 15 tiny beds of parsley here, mixed with onions. Parsnip of nine and a half beds, all good. Onions of 13 beds. Garlic, five beds. Carrots, seven beds. Radishes, one bed. Besides these there is also plenty of tarragon, sage, sedge, etc. Gooseberry planted in a row. Some peas. A few bushes of seed cabbage and two beds of seed parsley.”[57]

The 1721 description of the garden of György Bánffy’s mansion in Mezőörményes (Urmenisu de Campie, MS) mentions a popular contemporary garden structure, the gezebo, in addition to the plant species used:

“In this garden, to the right hand side, at the edge of the compartments extends southwards a row of fruit trees, pear and apple graft trees. In the middle is a shingled gazebo on four posts, standing on pedestals, surrounded by fourteen very fine vines, tied on stakes. Beside it, on the edge of the compartments, are fourteen vines on stakes. On the north side there are also some garden and wild rose trees on the edge of the compartments, and also very fine sage, sedge and white lilies…”[58]

As far as the registers are concerned, the economic diaries of the “grand dames” of the Principality, e.g., Anna Bornemisza, Zsuzsanna Lórántffy, or Mrs Sámuel Kálnoky [59,60,61], and the economic documents of the princely estates, e.g., of György Rákóczi I., are rich written sources of the Transylvanian aristocratic garden culture [62].

In the economic diary of Anna Bornemisza, the agricultural and horticultural crops of the princely estates are listed as sources of income. Figure 2a shows the income from wheat, barley, rye, oats, einkorn wheat, peas, lentils, and linseed of the Déva (Deva, HD) estate in 1667. Similar statements give an overview of the major crops grown on each estate at the time. Figure 2b is a list of expenditure on the mostly southern plants purchased by the princely court (oranges, lemons, pomegranates, olive trees, gooseberry, etc.) and the amounts spent on them, confirming the contemporary use of some exotic plants in Transylvanian gardens.

The detailed 99-page register of the manor of Fogaras [65], drawn up in 1632, lists the individual components of the estate as separate items, including the gardens (vegetable garden, orchard, deer garden), meadows, and fishponds relevant to our research. In a five-item list, the ‘lictariums’ are mentioned, the ancestors of modern-day marmalades, whose components also refer to the Transylvanian horticultural culture of the period (Figure 3). He also enumerates the garden tools (e.g., 6 iron hoes, 1 spade, 3 iron shovels, 1 three-pronged pitchfork, 1 two-pronged iron rake) and gives detailed descriptions of the individual gardens (e.g., the crops grown).

In the course of the work, a number of other archival sources were also researched. For reasons of space, we do not go into the detailed presentation and citation of these in this article, but only refer to them in the bibliography [66,67,68,69,70,71].

By their very nature, inventories show the assets primarily through the eyes of the compiler, which requires a cautious and critical approach and interpretation. Since the objects of the period (garden features) recorded in the inventory no longer exist, it is impossible to compare description with reality, and the situation is complicated by the inaccuracies and potential for misinterpretation arising from centuries of vocabulary, old-fashioned phrasing, contemporary terminology, and the difficulty of reading manuscript texts.

## 4. Results

The manor gardens of the period in question were of a mixed character, merging the concepts of the kitchen and ornamental gardens. If we classify the various garden types according to the plant species found in them, the gardens of the Late Renaissance period should be considered vegetable-flower gardens, geometrically compartmented gardens with some built elements. As early as the beginning of the sixteenth century, the compartment, in which the flowers were planted in regular order and with geometrical precision, became the central part of ornamental gardens throughout Europe. This garden motif, like many others, also appeared in Eastern Europe with a delay of a century. The distribution of the compartments was at once science and art, and horticultural handbooks taught in this era the design of the compartmented garden (Figure 4).

### 4.1. Definition of Late Renaissance Garden Units

We have defined a garden unit as a garden or garden section with distinct denomination and function (plant use). We have investigated and analyzed the frequency of occurrence and location of each garden unit. In the case of “Type A” sites, we have identified a total of three characteristic garden units on the basis of the archives, which occurred regularly in the examined Late Renaissance gardens: flower garden, vegetable garden, and orchard.

#### 4.1.1. The Flower Garden 

Mostly formal gardens planted with herbaceous flowers, often decorated with herbs, in regular order. Of the explored sites, 20 places are mentioned having flower gardens. Despite the fact that the flower garden was primarily decorative, it appears in many places together with kitchen gardens/allotments. 

“The design of the flower garden depends also closely on the composition of the landscape, and is the reflection of a lifestyle, a perspective, a philosophy and a changing socio-economic environment. With their flowers, the late Renaissance gardens of the Carpathian Basin were also the gardens of reality and freedom, because of the pomp of the West and the Ottoman dependency of the East. The symbol of national freedom at this time is the garden, where in addition to the flowers, the splendor and comfort of the gazebos showed this real world and the arising thoughts aof future independence as reconcilable,”[29]

As Csoma and Tüdős pointed out (see above), the garden must be approached as a microcosm of the landscape, and gardening must be regarded as the forerunner of landscape transformation.

We analyzed the inventories of the flower gardens in numerous cases thanks to the whole plant lists made of the species found there, but occasionally the species composition was not determined on the basis of live plants but from the prepared vegetable distillates. We collected 42 mentions of different flowers (with ornamental, medicinal, or condimental effects). The taxonomic identification of three of these flowers (marked with an asterisk in the Figure 5) has not been possible based on the folk nomenclature used in archival materials, so it is not known exactly what kind of flowers they are. The flower species used in Transylvanian Late Renaissance gardens are shown in Figure 5. These show that the most common flowers are rose (mentioned in 10 locations), sage, lily (nine locations), and carnation (seven locations), while some flowers, such as lilac, bellflower spur flower, etc., are found only in a single garden.

With regard to the varieties of the gardens, we found that the highest number of flower species was mentioned in the case of Komána (Comana de Jos, 25 different flower species) and Uzdiszentpéter (Sanpetru de Campie, 24 flower species). The number of described flower species largely depended on the season in which the census was taken and the depth of plant knowledge of the census taker.

#### 4.1.2. The Vegetable Garden

In general, a section of a geometrical garden was considered, mainly with ordered plantings of vegetables. Whenever one of the planted vegetables was in a larger proportion in the garden, the garden was named after the respective vegetable variety: cabbage garden in Görgényszentimre (Gurghiu, RO, 1652) or maize garden in Branyicska (Branisca, RO, 1757). Our research identified vegetable gardens on 30 sites based on the descriptions. In these 30 locations, we collected 30 mentions of different vegetables (and fodderplants). This highlights that the most common vegetable was the cabbage (mentioned in 20 locations), followed by some cereals and fodderplant species, e.g., wheat (20 locations), hemp (14 locations), and oats (13 locations), while some vegetables (such as pumpkin, chervil, asparagus etc.) were found only in a single garden (Figure 6).

#### 4.1.3. The Orchard

A garden area where mostly fruit trees were planted was considered. Similar to the vegetable garden, the name of the garden area could also be the name of the dominant fruit variety here: sour cherry garden in Uzdiszentpéter (Sânpetru de Câmpie, RO, 1679), apple garden in Csíkkozmás (Cozmeni, RO, 1688), plum tree garden in Görgényszentimre (Gurghiu, RO, 1652). Orchards are mentioned in 39 locations in the descriptions. Orchards (or fruit trees) were very often found in flower garden compartiments, too. This category includes the following sites: Négerfalva (Negrilesti, RO, 1697), Borberek (Vurpar, RO 1701), Szásznádas (Nadasul Sasesc, RO 1712), Szászcsanád (Cenade, RO 1736) Marosszentkirály (Sancraiu de Mures, RO, 1753)(B. Nagy, 1970), Sárpatak (Sapartoc, RO, 1736), Nagyercse (Ercea, RO, 1750), Vajdahunyad (Hunedoara, RO, 1681), Branyicska (Branisca, RO, 1726), Szentbenedek (Manastirea, RO, 1784), and Mezőörményes (Urmenis, RO, 1721). 

Figure 7 shows a terraced orchard garden on the castle hill from Segesvár (Sighisoara, RO), and some compartmented gardens organized in the manor courtyards (bottom, right), on the river shore, at the end of 17th century.

During the research, we found references to a total of 21 different fruit varieties in 39 residence gardens. The fruit varieties mentioned in contemporary inventariums and their frequency are shown in Figure 8 and Table 2 for each location. These show that the most popular fruits are plums (mentioned in 23 locations), grapes (19 locations), and sour cherries (17 locations). According to records, the rarest fruits are rowan, quince, cherry, almond, and raspberry. At the same time, Mediterranean plants are also included in the inventarium at two locations: lemon in Uzdiszentpéter and olive tree in Fogaras.

## 5. Discussion

In the garden history of Transylvania, the Late Renaissance style was almost one hundred years late in spreading compared with other parts of Europe. In spite of that, in terms of the number of gardens in Transylvania, the Renaissance can be called the leading garden style in this part of the country. The number of Late Renaissance gardens is much higher than the number of Early Renaissance or later Baroque Gardens. This is a consequence of the political and economic independence of Transylvania in the 17th century. 

The history of the Transylvanian Late Renaissance garden is mainly about economical sustainability. This is why mixed gardens (ornamental and vegetable gardens and even orchards together) frequently appear, although in many places various kinds of kitchen and vegetable gardens or orchards were also independently represented. 

In terms of their compositional characteristics, Transylvanian Late Renaissance gardens are largely integrated into the general European system. They followed the models, sometimes as simplified small-scale paraphrases of them, but there are also examples that have specific local characteristics (in ethnographic and topographic terms or in plant use). 

Given the state of what is left over from these historical artefacts, restoration in the strict sense is impossible. Devastation, missing archival sources, changing ownership, and sustainability reasons make the restoration work even harder. During the investigation, analysis, and fieldwork regarding the Transylvanian ensembles, we had ample contact with local stakeholders, politicians, owners, NGOs, users, and other people related to the Transylvanian ensembles. The core of the problem is concentrated around two poles: one of heritage and cultural meaning, the other centered on the search for new functions and uses. These two are often contradictory and conflicting; they can be categorized in the polarity between development and conservation. In all landscape architectural projects, this contradiction plays a role, but in the case of historical phenomena they are even more pronounced and demand special attention. 

It will be a major challenge for landscape architecture to take into account the historical values and to integrate them with new functions and uses as well as the recent demands of improving water management, energy transition, and the creation of comfortable and healthy living environments for people.

## 6. Conclusions

This research offers an overview of the four-hundred-year history of the Late Renaissance gardens of Transylvania, focusing on the plant use of the Late Renaissance manor gardens. The research has collected and ordered the most important gardens of that era, highlighting the most representative flower, vegetable, and fruit species cultivated and described in archival documents. 

Moreover, the study defines the most typical garden types of the study area in the period concerned, naming the compartmented flower gardens, vegetables gardens, and orchards as main garden types. However, the study also shows that the different types of gardens were very often mixed, with flower gardens often containing vegetable or fruit trees, and vice versa.

The plant lists and registers that have been discovered help us to gain an insight into the plant use habits of the Transylvanian manor gardens of the 17th and 18th centuries and demonstrate the sophistication of the garden culture of the period and the richness of the plant selection used. The research is of horticultural and landscape architectural importance as it highlights the plant use traditions and demonstrates the continuity and applicability of many cultivated plants in Transylvanian gardens.

## Figures and Tables

**Figure 1 plants-12-01798-f001:**
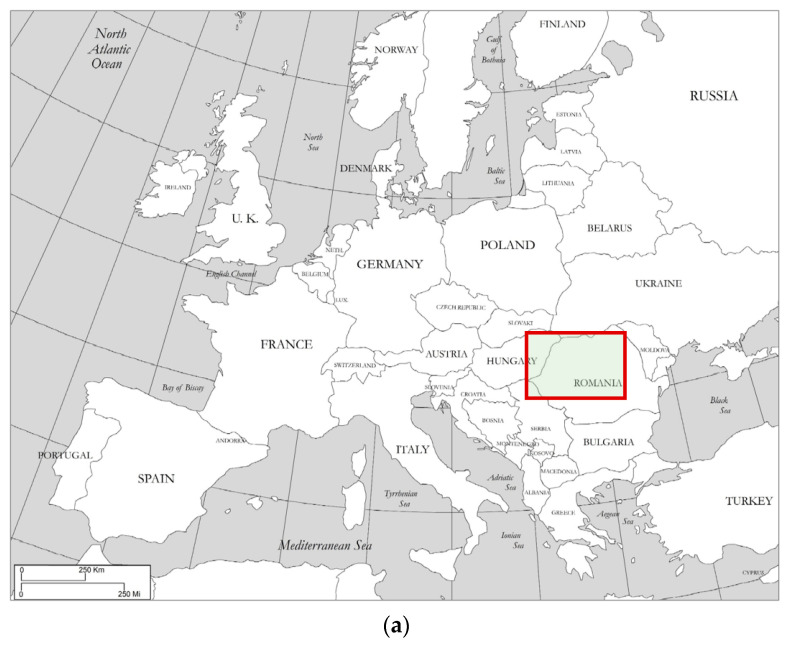
(**a**) The study area on the map of Europe (Source: Prepared by Authors). (**b**) Spatial distribution of the investigated sites on the map of Transylvania (Source: Prepared by Authors).

**Figure 2 plants-12-01798-f002:**
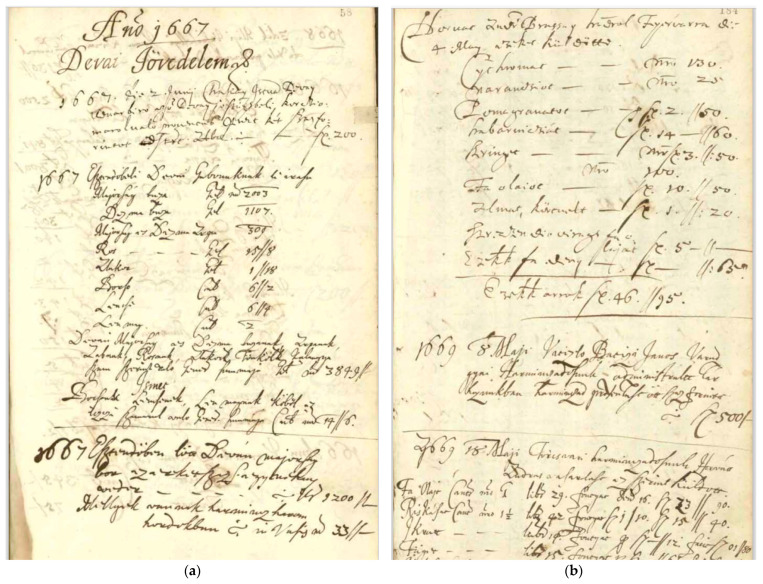
(**a**) List of cereals and fodder plants from Déva (Deva, HD), 1667. (Source: [63]), (**b**) List of southern plants purchased by the princely court in Fogaras, 1667 (Source: [64]).

**Figure 3 plants-12-01798-f003:**
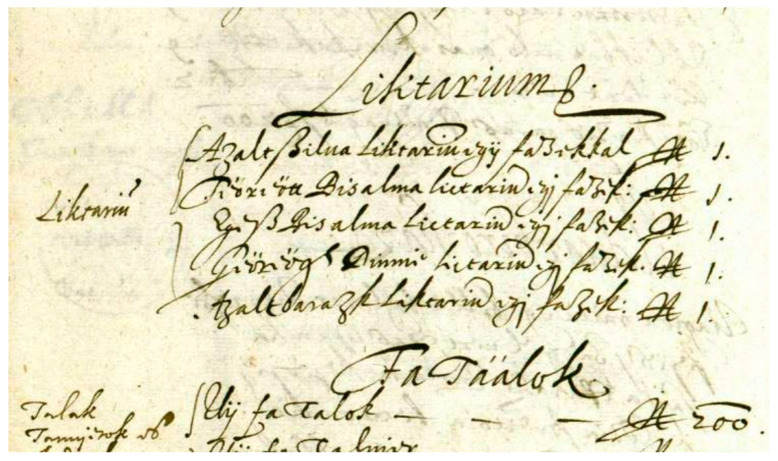
List of ‘lictariums’ in Fogaras manor estate, from 17th century (Source: [66]).

**Figure 4 plants-12-01798-f004:**
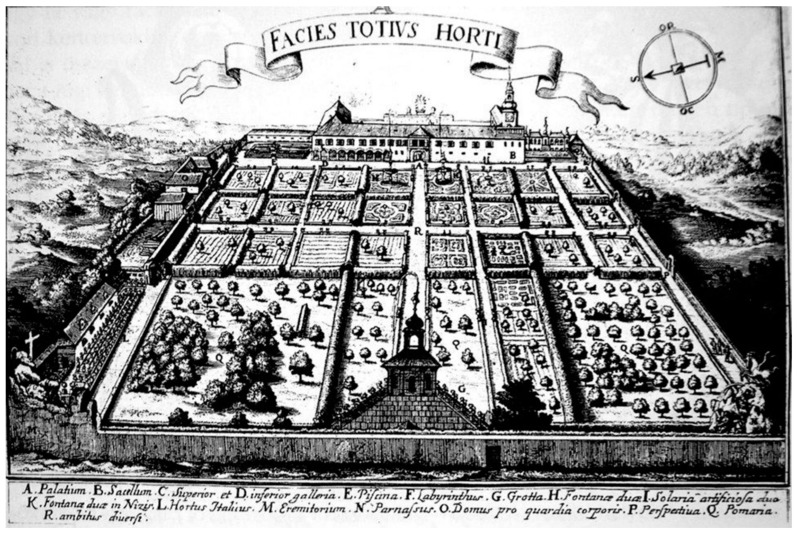
Drawing of the Bishop’s Garden from of Bratislava in 1664, showing the compartmented division of planted beds (Source: [26]).

**Figure 5 plants-12-01798-f005:**
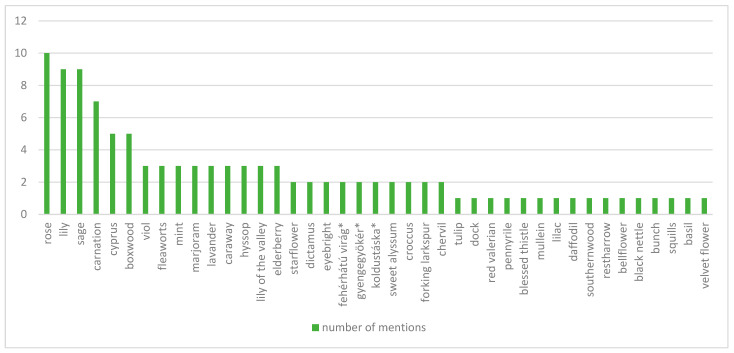
Flower species used during 17–18th centuries in the Transylvanian residential gardens, mentioned in the inventories and other archival materials. The taxonomic identification of the species marked with an asterisk has not been possible based on the folk nomenclature used in archival materials, so it is not known exactly what kind of flowers they are. (Source: prepared by the Authors, based on [45,46,47,48,49,50,51,52,53,54,55,56,57,58,59,60,61,62,63,64,65,66,67,68,69,70,71]).

**Figure 6 plants-12-01798-f006:**
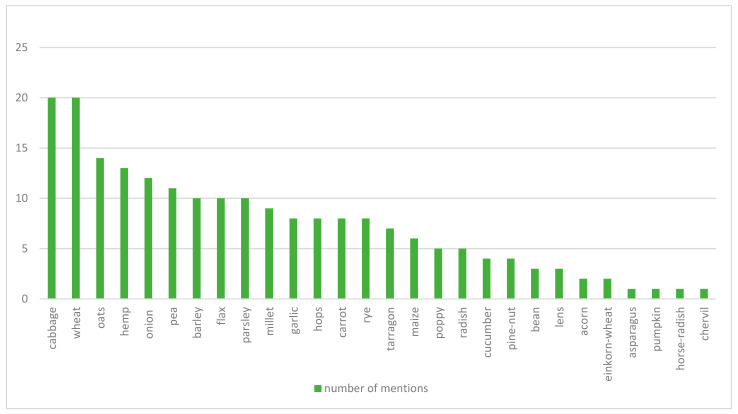
Frequency of the most common vegetables, cereals and fodder plants used during the 17–18th centuries in the Transylvanian residential gardens, mentioned in the inventories and other archival materials (Source: prepared by the Authors, based on [45,46,47,48,49,50,51,52,53,54,55,56,57,58,59,60,61,62,63,64,65,66,67,68,69,70,71]).

**Figure 7 plants-12-01798-f007:**
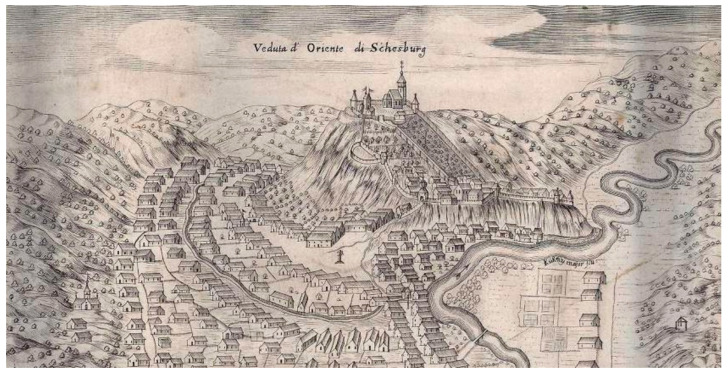
View of Schassburg the turn of 17/18 centuries, with representation of orchards and compartmented gardens (Source: [72]).

**Figure 8 plants-12-01798-f008:**
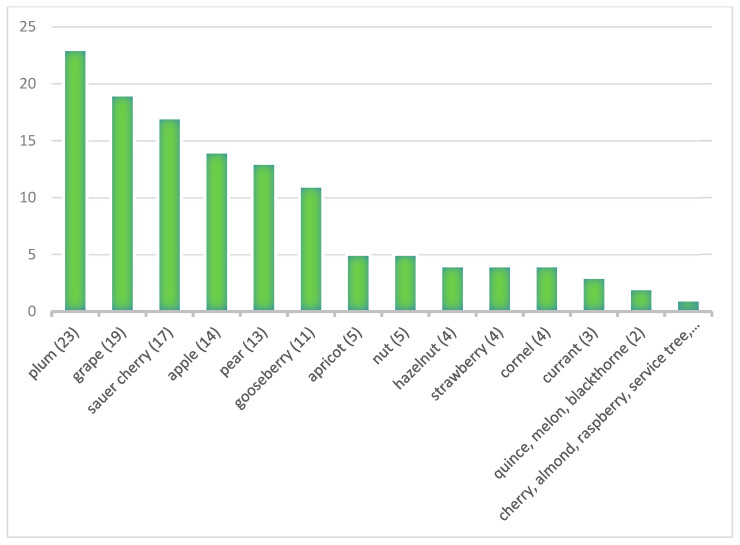
Fruit species mentioned in archival materials and its frequency in the gardens (Source: prepared by the Authors, based on [45,46,47,48,49,50,51,52,53,54,55,56,57,58,59,60,61,62,63,64,65,66,67,68,69,70,71]).

**Table 1 plants-12-01798-t001:** List of during the research identified Renaissance garden’s locations from Transylvania. (Source: prepared by Authors).

No	Locations with Garden Description (Type “A”) Hungarian Name/Romanian Name	Data (Year)	No	Locations without Garden Description, Only Mentioning the Existence of the Garden (Type “B”) Hungarian Name/Romanian Name
1	Kisbarcsa/Barcea Mica	1624	51	Porumbák/Porumbac
2	Fogaras/Fagaras	1632	52	Aranyosmeggyes/Mediesul Aurit
3	Siménfalva/Simonesti	1636	53	Lugos/Lugoj
4	Tasnád/Tasnad	1644	54	Lippa/Lipova
5	Nagyteremi/Tirimia	1647	55	Odvos/Odvos
6	Királyfalva/Craiesti	1647	56	Marosillye/Ilia
7	Meggykerék/Mescreac	1647	57	Szászsebes/Sebes
8	Drassó/Drasov	1647	58	Algyógy/Geoagiu
9	Marosvécs/Brancovenesti	1648	59	SzászcsanádCenade
10	Komána/Comana de Jos	1648	60	Sorostély/Sorostin
11	Sajókeresztúr/Cristesti	1648	61	Alsóárpás/Arpasu de Jos
12	Görgényszentimre/Gurghiu	1652	62	Sáros/Soars
13	Gerend/Luncani	1652	63	Kézdiszentlélek/Sanzieni
14	Magyarbükkös/Bichis	1655	64	Pálos/Palos
15	Búzábocsárd/Bucerdea Granoasa	1658	65	Bögöz/Mugeni
16	Mezőszengyel/Sanger	1656	66	Sárpatak/Sarpotoc
17	Szurdok/Surduc	1657	67	Keresd/Cris
18	Bethlen/Beclean	1661	68	Martonfalva/Metis
19	Déva/Deva	1667	69	Szásznádas/Nades
20	Mezőbodon/Papiu Ilarian	1679	70	Szentdemeter/Dumitreni
21	Nagysajó/Comuna Sieu	1681	71	Nagyercse/Ercea
22	Oprakercisóra/Cartisoara	1683	72	Mezőzáh/Zaul de Campie
23	Nyujtód/Lunga	1684	73	Paszmos/Posmus
24	Gernyeszeg/Gornesti	1685	74	Kentelke/Chintelnic
25	Csíkkozmás/Cozmeni	1688	75	Búza/Búza
26	Nagybún/Boiu Mare	1692	76	Gyeke/Geaca
27	Borberek/Vurpar	1694	77	Kóródszenmárton/Coroisanm
28	Kővár/Cetatea Chioarului	1694	78	Kendilóna/Luna de Jos
29	Vajdahunyad/Hunedoara	1695	79	Négerfalva/Negrilesti
30	Alvinc/Vintul de Jos	1696	80	Szamosfalva/Somesen-Cluj
31	Szentbenedek/Manastireni	1696	81	Belényes/Beius
32	Miklósvár/Miclosora	1698	82	Szilágysomlyó/Simleul Silvaniei
33	Egeres/Aghires	1699	83	Nagybánya/Baia Mare
34	Zentelke/Sancraiu	1715	84	Halmi/Halmeu
35	Malomvíz/Grid	1716	85	Küküllővár/Cetatea de Balta
36	Mezőörményes/Urmenis	1721	86	Székelyhíd/Sacueni
37	Koronka/Corunca	1724		
38	Marosszentkirály/Sancraiu de Mures	1725		
39	Aranykút/Aruncuta	1728		
40	Kaplyony/Coplean	1729		
41	Bonchida/Bontida	1736		
42	Branyicska/Branisca	1757		
43	Szilágycsehi/Cehu Silvaniei	17. c.		
44	Gyulafehérvár/Alba Iulia	17. c.		
45	Ebesfalva/Daumbraveni	17. c.		
46	Ádámos/Adamus	17. c.		
47	Olasztelek/Talisoara	17. c.		
48	Radnót/Iernut	17. c.		
49	Sepsiköröspatak/Valea Crisului	17. c.		
50	Uzdiszentpéter/Sanpetru de Campie	17. c.		

**Table 2 plants-12-01798-t002:** List of residential gardens with fruit gardens with a specification of the used fruit varieties (Source: prepared by the Authors, based on [45,46,47,48,49,50,51,52,53,54,55,56,57,58,59,60,61,62,63,64,65,66,67,68,69,70,71]).

No	Location	Data (Year)	Apple	Apricot	Sorb	Quince	Lemon	Melon	Nut	Strawberry	Gooseberry	Cherry	Blackthorne	Pear	Almond	Raspberry	Sauercherry	Hazelnut	Oil Tree	Cornel	Plum	Grape	Currant
1	Kisbarcsa	1624																				**x**	
2	Fogaras	1632			x	x					x								**x**		x	x	
3	Siménfalva	1636															**x**				**x**	**x**	
4	Tasnád	1644	x											x						x	**x**	**x**	
5	Nagyteremi	1647																				**x**	
6	Királyfalva	1647																					
7	Meggykerék	1647																					
8	Drassó	1647															**x**						
9	Marosvécs	1648	**x**											**x**			**x**			**x**	**x**	**x**	
10	Komána	1648		**x**					**x**	**x**		**x**	**x**				**x**			**x**	**x**		
11	Görgény	1652												**x**							**x**		
12	Gerend	1652															**x**				**x**	**x**	x
13	Búzábocsárd	1658																				**x**	
14	Mezőszengyel	1656																					
15	Szurdok	1657															**x**						
16	Bethlen	1661																			**x**	**x**	
17	Mezőbodon	1679	**x**								**x**										**x**		
18	Uzdiszentpéter	1679		**x**			**x**	**x**	**x**	**x**	**x**		**x**							**x**	**x**		
19	Nagysajó	1681	**x**						**x**	**x**	**x**			**x**			**x**	**x**			**x**	**x**	**x**
20	Oprakercisóra	1683																			**x**		
21	Csíkkozmás	1688	**x**								**x**			**x**									
22	Nagybún	1692	**x**											**x**			**x**				**x**		
23	Borberek	1694	**x**	**x**					**x**	**x**				**x**			**x**	**x**				**x**	
24	Kővár	1694																				**x**	
25	Vajdahunyad	1695																				**x**	
26	Szentbenedek	1696	**x**								**x**			**x**			**x**				**x**	**x**	
27	Egeres	1699	**x**								**x**			**x**			**x**	**x**			**x**		
28	Zentelke	1715															**x**						
29	Grid	1716	**x**																		**x**		
30	Mezőörményes	1721	**x**	**x**					**x**		**x**				**x**		**x**	**x**			**x**	**x**	
31	Koronka	1724		**x**										**x**							**x**		
32	Marosszentkirály	1725															**x**					**x**	
33	Aranykút	1728	**x**			**x**		**x**			**x**			**x**		**x**	**x**				**x**	**x**	
34	Bonchida	1736									**x**						**x**				**x**	**x**	
35	Gernyeszeg	1751	**x**											**x**							**x**		
36	Branyicska	1757	**x**														**x**				**x**	**x**	
37	Szilágycsehi	17. c.																			**x**		
38	Gyulafehérvár	17. c.	**x**											x									
39	Ebesfalva	17. c.									**x**												x
	**TOTAL Number**		**14**	**5**	**1**	**2**	**1**	**2**	**5**	**4**	**11**	**1**	**2**	**13**	**1**	**1**	**17**	**4**	**1**	**4**	**23**	**19**	**3**

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
