# Peer review of "Plant Use in the Late Renaissance Gardens of the 17–18th Century Transylvania"

_plants, 2023, doi:10.3390/plants12091798_

Round 1

Reviewer 1 Report

Dear Authors,

I find this manuscript interesting as it focuses on findings  to define and locate the most frequent occurrences of late Renaissance gardens of the 17-18 th century in Transylvania.

Objectives:

At the end of objectives you should formulate the research question of the study.

3. Methodology

In the methodology section, you should mention some details about the archives (e.g. if you used archives from both Romania and Hungary, the duration of the archive study).

Moreover, you should mention the stage of documentation.

When you referred to the actual aspect of the late Renaissance gardens art you mentioned that: "As far as Transylvanian Late Renaissance Garden art is concerned, the material heritage is rather scarce. None of the Transylvanian castle gardens survive in their original form, and very few elements of Renaissance garden art have survived. What has survived is mainly associated with large-scale earthworks, such as those related to water features (canals, moats, fish ponds), terraces (e.g. orchards on hillsides)." (Lines 111-115). Did you obtain this information during filed campaigns? Can you mention this detail also.

4. Discussion and Results

This section should be renamed Results and discussion.

In this section only the results are presented. I missed the discussion section.

For the development of discussion section, you should present critically the results synthetically and how they can be interpreted from the perspective of previous studies that emphasize similarities or differentiations, especially in studies dedicate to the gardens located in Central or Eastern European countries. Relate your findings to previous studies.

Moreover, in the discussion section, you should refer to:

(1) the importance of the study from the methodologically  and scientifically point of view

(2) the limitations of the study and future research;

Minor comments:

To add word source to the caption of figure 4.

Author Response

Authors reply to Reviewer no 1

We would like to thank the reviewers for their reviews, suggestions and remarks. It has greatly improved the content, form and text.

We reworked the text integrating the specific remarks of all three reviewers and we have followed the ratings of the criteria: Reviewer 1 and Reviewer 2

Point 1:

In the methodology section, you should mention some details about the archives (e.g. if you used archives from both Romania and Hungary, the duration of the archive study).

Response 1:

We added the requested details to the text.

Point 2:

Moreover, you should mention the stage of documentation.

When you referred to the actual aspect of the late Renaissance gardens art you mentioned that: "As far as Transylvanian Late Renaissance Garden art is concerned, the material heritage is rather scarce. None of the Transylvanian castle gardens survive in their original form, and very few elements of Renaissance garden art have survived. What has survived is mainly associated with large-scale earthworks, such as those related to water features (canals, moats, fish ponds), terraces (e.g. orchards on hillsides)." (Lines 111-115). Did you obtain this information during filed campaigns? Can you mention this detail also.

Response 2:

We added the requested details to the text.

Point 3:

  1. Discussion and Results

This section should be renamed Results and discussion.

In this section only the results are presented. I missed the discussion section.

For the development of discussion section, you should present critically the results synthetically and how they can be interpreted from the perspective of previous studies that emphasize similarities or differentiations, especially in studies dedicate to the gardens located in Central or Eastern European countries. Relate your findings to previous studies.

Moreover, in the discussion section, you should refer to:

(1) the importance of the study from the methodologically and scientifically point of view

(2) the limitations of the study and future research;

Response 3:

Done: we created a chapter „4.Results” and a chapter „5.Discussions”

Point 4:

Minor comments:

To add word source to the caption of figure 4.

Response 4: Done

Reviewer 2 Report

An interesting and well-written paper.

A general comment is about plant names. Why are plant names not given as Linnaean binomials? Is this because using Linnaean nomenclature would give a false precision, specially for plants whose name in the paper left me confused: see detailed comments.

Detailed comments.

Line 83. ‘analysis’, not ‘analyzis’

Line 90. ‘of the Carpathian’.

Figure 1b. excellent map.

Line 111. Clarify ‘art’; is this the same as ‘pictorial sources’ on p. 123.

Line 161. Clarify ‘structure’ and ‘structures’.

Line 219. Should ‘Discussion and results’ be ‘Results and discussion’?

Lines 254-256. Good.

Line 275, ‘plantings’ not ‘plantation’

Line 284. ‘turbot’ is English for an edible flatfish, Scopthalamus maximus.

Figure 5. What is ‘southernwood’?

Figure 6. Does ‘acorn’ mean ‘oak’, and ‘pine-nut’ mean ‘pine’?

Line 286. ‘fodder’, not ‘fodden’,

Figure 8. What kind of plant is a ‘sorb’? What is meant by ‘oil’?

Line 317. ‘used’, not ‘usde’.

Author Response

Authors reply to Reviewer no 2

We would like to thank the reviewers for their reviews, suggestions and remarks. It has greatly improved the content, form and text.

We reworked the text integrating the specific remarks of all three reviewers and we have followed the ratings of the criteria: Reviewer 1 and Reviewer 2

Point 1:

A general comment is about plant names. Why are plant names not given as Linnaean binomials? Is this because using Linnaean nomenclature would give a false precision, specially for plants whose name in the paper left me confused: see detailed comments.

Answer 1:

Plant names in old texts do not always refer to the exact species name. Often only the genus can be determined from the mentions. This is the reason why we didn’t used the Linnaean binomials. Anyway, we improved a bit based on discussions with some botanists.

Point 2:

Detailed comments – and answers

Line 83. ‘analysis’, not ‘analyzis’ - corrected

Line 90. ‘of the Carpathian’ - corrected

Figure 1b. excellent map – thank you

Line 111. Clarify ‘art’; is this the same as ‘pictorial sources’ on p. 123.

Answer:

The ’art’ is about everything artistical related tot he garden: composition, colour, layout and spatial structure etc.

The ’pictorial sources’ are visual materials (archives) helping us to understand the gardens, showing it as a whole or ome deatails of it, like: paintings, drawings, gravures, pictures, photographs, sketches, designs, plans, sections, elevations, maps etc.

Line 161. Clarify ‘structure’ and ‘structures’. - corrected

Line 219. Should ‘Discussion and results’ be ‘Results and discussion’?

Answer:

We improved this section and we divided it in two different chapters: „4. Results” and „5. Discussions”

Lines 254-256. Good.

Line 275, ‘plantings’ not ‘plantation’ - corrected

Line 284. ‘turbot’ is English for an edible flatfish, Scopthalamus maximus.  – we corrected itt o „chervil”

Figure 5. What is ‘southernwood’?  - Artemisia abrotanum

Figure 6. Does ‘acorn’ mean ‘oak’, and ‘pine-nut’ mean ‘pine’?

Answer:

acorn is the oaknut (the crop of an oak)

pine-nut is the edible sees of a pine

Line 286. ‘fodder’, not ‘fodden’, - corrected everywhere

Figure 8. What kind of plant is a ‘sorb’? What is meant by ‘oil’?

We corrected the oil-tree to olive (Olea europaea)

We corrected the „sorb” to Service tree (Sorbus aucuparia) used more often in the everyday conversation

Line 317. ‘used’, not ‘usde’.

corrected